# Chemo-Sensitization of CD133+ Cancer Stem Cell Enhances the Effect of Mesenchymal Stem Cell Expressing TRAIL in Non-Small Cell Lung Cancer Cell Lines

**DOI:** 10.3390/biology10111103

**Published:** 2021-10-26

**Authors:** Kamal Shaik Fakiruddin, Moon Nian Lim, Norshariza Nordin, Rozita Rosli, Syahril Abdullah

**Affiliations:** 1Haematology Unit, Cancer Research Centre, Institute for Medical Research (IMR), National Institutes of Health (NIH), Ministry of Health Malaysia, Shah Alam 40170, Malaysia; limmn@moh.gov.my; 2UPM-MAKNA Cancer Research Laboratory, Institute of Bioscience, Universiti Putra Malaysia, Seri Kembangan 43400, Malaysia; rozita@upm.edu.my (R.R.); syahril@upm.edu.my (S.A.); 3Medical Genetics Laboratory, Department of Biomedical Sciences, Faculty of Medicine & Health Sciences, Universiti Putra Malaysia, Seri Kembangan 43400, Malaysia; shariza@upm.edu.my; 4Genetics and Regenerative Medicine Research Group, Faculty of Medicine & Health Sciences, Universiti Putra Malaysia, Seri Kembangan 43400, Malaysia

**Keywords:** non-small cell lung cancer, cancer stem cell, chemo-sensitization, first-line chemotherapies, mesenchymal stem cell, TRAIL

## Abstract

**Simple Summary:**

The anti-tumor properties of mesenchymal stem cell (MSCs) expressing TNF-related apoptosis inducing ligand (TRAIL) or MSC-TRAIL have been well documented by several reports. However, some tumors are resistant to TRAIL due to the existence of cancer stem cells (CSCs). Chemo-sensitization of tumors and their CSCs has been reported to enhance TRAIL-mediated inhibition. In this study, we examined the effect of pre-treatment using first-line chemotherapies on MSC-TRAIL-induced inhibition in non-small cell lung cancers (NSCLCs)–derived CSCs. We found that these chemotherapies were able to induce a chemo-sensitization effect to the CSC, thus improving the MSC-TRAIL-induced inhibition. We also noticed that the effect of chemo-sensitization was cell type specific and selecting chemotherapies for the right NSCLC subtypes might help in inducing a more meaningful combinatory effect. As such, this study has proven that chemo-sensitization of the CSCs was able to enhance the MSC-TRAIL-induced inhibition in NSCLC cell lines.

**Abstract:**

Pre-clinical studies have demonstrated the efficacy of mesenchymal stem cells (MSCs) expressing tumor necrosis factor (TNF)-related apoptosis-inducing ligand (TRAIL) or MSC-TRAIL against several tumors. However, due to the existence of cancer stem cells (CSCs), some tumors, including non-small cell lung cancer (NSCLC), exhibit TRAIL resistance. This study was designed to evaluate the capacity of using first-line chemotherapies including cisplatin, 5-fluorouracil (5-FU) and vinorelbine to act as a chemo-sensitizer on CD133+ (prominin-1 positive) CSCs derived from NSCLC cell lines (A549, H460 and H2170) for the purpose of MSC-TRAIL-induced inhibition. We showed that MSC-TRAIL was resistant to all three chemotherapies compared to the NSCLC cell lines, suggesting that the chemotherapies had little effect on MSC-TRAIL viability. Pre-treatment using either cisplatin or 5-FU, but not with vinorelbine, was able to increase the efficacy of MSC-TRAIL to kill the TRAIL-resistant A549-derived CSCs. The study also demonstrated that both 5-FU and vinorelbine were an effective chemo-sensitizer, used to increase the anti-tumor effect of MSC-TRAIL against H460- and H2170-derived CSCs. Furthermore, pre-treatment using cisplatin was noted to enhance the effect of MSC-TRAIL in H460-derived CSCs; however, this effect was not detected in the H2170-derived CSCs. These findings suggest that a pre-treatment using certain chemotherapies in NSCLC could enhance the anti-tumor effect of MSC-TRAIL to target the CSCs, and therefore the combination of chemotherapies and MSC-TRAIL may serve as a novel approach for the treatment of NSCLC.

## 1. Introduction

Lung cancer is an uncontrolled growth of malignant cells that can occur in any part of the lungs [1]. These malignant cells do not have the function of normal lung cells and are able to metastasize into different parts of the body and organs [2]. The majority of lung cancer is detected as non-small cell lung cancer (NSCLC), which accounts for 85% of all cases, whereas the other 15% are detected as small cells [3]. Lung cancer is the major cause of mortality and morbidity worldwide, killing more than 1.7 million people annually [3]. Despite current advances in cancer therapy, many of the treatments given to lung cancer patients are still unable to completely cure the disease [4]. This is due to the existence of lung cancer CSCs that are spared by chemotherapies [5]. Chemoresistance is the main characteristic and the hallmark that differentiates CSCs from non-CSCs, as most chemotherapies are only able to target non-CSCs, leaving CSCs to survive, which will eventually lead to tumor repopulation [6].

Tumor necrosis factor-related apoptosis-inducing ligand (TRAIL) (also known as APO-2L) is one of several members of the TNF gene superfamily that induces apoptosis. TRAIL activates the extrinsic apoptosis pathway through binding to its two specific agonistic receptors, known as DR4 and DR5, and three antagonistic decoy receptors, known as DcR1, DcR2 and osteoprotegerin (OPG) [7]. The anti-tumor activity of TRAIL has been shown in several preclinical tumor models, including colorectal cancer [8], glioblastoma [9] and NSCLC [10]. However, TRAIL-resistance can also be seen in several other tumor models, including breast cancer [11], as well as ovarian [12] and pancreatic cancer [13]. The dysregulation and evasion of CSCs to apoptosis could be the main factor contributing to TRAIL resistance in some tumors [14]. Thus, sensitizing the CSCs and tumors to apoptosis could provide a way to enhance the overall sensitivity of the tumor to TRAIL-induced apoptosis [15,16].

Several limitations, including its short half-life and poor bioavailability, have hindered the translation of TRAIL into the clinic [17,18]. Owing to its small molecular weight, systemically delivered TRAIL is mostly excreted by the kidneys through renal filtration [19]. Successful attempts have been made to increase the serum bioavailability of TRAIL through the addition of isoleucine zipper [20], N-terminus his-tagged protein [21] or immunoglobulin chain [22] into its structure. However, the addition of his-tag or isoleucine zipper is potentially immunogenic [23] and may contribute to hepatotoxicity compared to the native TRAIL [24]. The current developments in cytotherapy, using cells as a vehicle for therapeutic agents, could serve as the most effective approach for the continuous delivery of TRAIL to the target site [25].

Mesenchymal stem cells or MSCs are adult multipotent stem cells that can be derived from different sources, including adipose tissue, the umbilical cord and bone marrow [26]. The ability of MSCs to home and integrate into the tumor environment have expanded the scope of treatments using these cells, not only in degenerative diseases, but also as a vehicle for the delivery of therapeutic agents to treat cancer [27]. MSCs have been used as a delivery system for cytokines, including interleukins [28,29], interferon [30] and pro-apoptotic proteins such as TRAIL [31,32,33,34,35] in several preclinical models. However, compared to other anti-tumor agents that have utilised MSCs as a delivery system, only MSCs expressing TRAIL (MSC-TRAIL) have been reported to be effective against several tumors with minimal toxicity [36].

The synergistic anti-tumor activity of the combination of chemotherapies and MSC-TRAIL seen in several tumor models suggests that this combination could be an effective anti-cancer therapy [37,38,39]. The dual effects of common chemotherapies in cancer, either as a cytotoxic drug or as a sensitizer to the effect of TRAIL and MSC-TRAIL have also been demonstrated in several tumor xenografts including those of brain cancer [40], hepatocellular carcinoma [41], breast cancer [42] and NSCLC [43]. Furthermore, TRAIL and MSC-TRAIL either alone or in combination with other chemotherapies have also been shown to be effective in targeting CSCs in breast [44,45] and liver cancer [46]. Although we have found that MSC-TRAIL was effective in targeting the cell lines and their CSCs, one out of the three NSCLC cell lines used was TRAIL-resistant [47]. Thus, as studies have suggested the benefit of combining other chemotherapies and TRAIL, we postulated that a pre-treatment using first-line chemotherapies such as cisplatin, 5-fluorouracil (5-FU) or vinorelbine, as a chemo-sensitizer, would lead to greater sensitivity of tumor cells and its CSCs to MSC-TRAIL-induced apoptosis, particularly in the TRAIL-resistant cells.

## 2. Materials and Methods

### 2.1. Culture of Human Adipose-Derived Mesenchymal Stem Cells

Human adipose-derived mesenchymal stem cells (MSCs; cat no: ATCC^®^ PCS-500-011©) were purchased from the American Type Culture Collection (ATCC, Manassas, VA, USA). The cells were cultured in specific growth medium containing knockout Dulbecco’s modified Eagle medium (DMEM-KO), 1% penicillin/streptomycin, 2 mM of l X glutamine (200 mM stock), 10% fetal bovine serum (FBS), 5 ng/mL fibroblast growth factor (FGF) basic and 5 ng/mL recombinant epidermal growth factor (rhEGF).

### 2.2. Culture of NSCLC Lines

Three types of human non-small cell lung cancer cell lines (H2170, A549, and H460) were used in this study. The lung squamous cell carcinoma cell line (H2170) (cat no: ATCC^®^ CRL-5928) was purchased from the American Type Culture Collection (ATCC, Manassas, VA, USA), whereas the other two cell lines (both constitutively expressing luciferase), human adenocarcinoma (A549) (cat no: JCRB1414) and large cell lung cancer (H460) (cat no: JCRB1407), were purchased from Cell Bank Australia (Westmead, NSW, Australia). The A549 cell line was cultured in Roswell Park Memorial Institute-1640 (RPMI)-1640 complete medium containing 1% penicillin/streptomycin, 1X non-essential amino-acid solution and 10% heat-inactivated FBS. For the H460 cell line, a complete medium was prepared by adding 15% heat-inactivated FBS to RPMI-1640 medium containing 1% penicillin/streptomycin and 0.08 µg/mL insulin (4 mg/mL stock). H2170 complete medium was prepared by adding 10% FBS and 1% penicillin/streptomycin to RPMI-1640. The cells were maintained in 75 cm^2^ flasks (Nunc, Thermo Fisher Scientific, Inc., Waltham, MA, USA) and harvested using 0.25% trypsin–ethylenediaminetetraacetic acid (EDTA) when the cells reached 80% confluence. All cells were grown at 37 °C in a humidified atmosphere of 5% CO_2_. All culture reagents were obtained from Gibco (Thermo Fisher Scientific, Inc., Waltham, MA, USA).

### 2.3. Production of MSC-TRAIL

The mesenchymal stem cells were transduced with the human full-length TRAIL gene (NM_003810.2) encoding the membrane-bound TRAIL tagged with a red fluorescence protein (mCherry) using lentivirus. The MSC-TRAIL was successfully characterised based on TRAIL expression and multipotent characteristics (adipogenesis, chondrogenesis and osteogenesis), as shown in our previous report [47].

### 2.4. Isolation and Characterisation of CD133+ CSCs

The CD133+ population from all of the NSCLC cell lines was isolated and successfully characterised based on clonogenicity, sphere formation and aldehyde dehydrogenase expression [47]. In brief, the NSCLC cell lines were harvested, washed and stained with the CD133 (prominin-1) antibody, 1:10 dilution (Clone: AC133; Isotype: Mouse IgG1 kappa) (Miltenyi Biotec, Bergisch Gladbach, Germany) and incubated for 15 min in dark. Stained cells were then washed using DPBS, precipitated and subsequently resuspended in ice-cold DPBS with 2% FBS before being subjected to specific CD133+ (cancer stem cells, CSCs) and CD133− (non-CSCs) isolation using a fluorescence-activated cell sorter (FACSAria III; BD Biosciences, San Jose, CA, USA).

### 2.5. Analysis of Population Doubling Time

The cumulative population doublings (CPD) and population doubling time (PDT) in all the MSCs (MSC-WT (wild-type), MSC-EV (empty vector) and MSC-TRAIL) and NSCLC cell lines (A549, H460 and H2170) were analysed by comparing the number of cells harvested at every 72 h to the initial number of cells seeded (2.5 × 10^5^ cells in 2 mL of a 6-well plate) using formulas as below:CPD = [Log10 (H) − Log10 (I)]/Log10 (2)
PDT = [(No. of days in culture x 24 h)]/CPD

H—No. of cells harvested

I—No. of cells seeded (2.5 × 10^5^ cells)

### 2.6. Analysis of IC_50_ Values of Different Chemotherapies

The IC_50_ values of different chemotherapies (cisplatin, 5-fluorouracil/FU and vinorelbine) used for the treatment of NSCLC were determined in the NSCLC cell lines (A549, H460 and H2170) and MSCs (MSC-WT (wild-type), MSC-EV (empty vector) and MSC-TRAIL) using a proliferation/MTS assay. Cells were seeded in a 96-well plate (5.0 × 10^3^ in 50 µL complete medium) and grown overnight prior to the addition of chemotherapeutic drugs. To produce treatment stocks consisting of 10 mM of cisplatin and vinorelbine tartrate and 0.25 M of 5-FU, different volumes of solvents were added into each of the drugs with different weights calculated, according to their molecular weight. Different concentrations of drugs ((200 µM, 100 µM, 50 µM, 25 µM, 12.5 µM, 6.25 µM, 3 µM for cisplatin and vinorelbine) and 40 mM, 20 mM, 10 mM, 5 mM, 2.5 mM, 1.25 mM and 0.6 mM for 5-FU) were added into each of the wells containing the tumor cells in triplicate. After 48 h of treatment, 10 µL of MTS was added into each well and incubated for another 3 h. The samples were then subjected to absorbance reading at 490 nm using a plate reader (Envision, Perkin Elmer, Waltham, MA, USA). Cell proliferation was calculated according to the following formula: cell proliferation (%) = [absorbance (cells with treatment)/absorbance (cells without treatment)] × 100. The IC_50_ values of the chemotherapeutic drugs for each of the cell lines were calculated using a linear regression formula (y = mx + c) from a scatter plot, wherein the x-axis represents the concentration of drugs (in log10), the y-axis expresses the percentage of cell viability, “m” is the gradient, and “c” is the y-intercept value. The IC_50_ values of the drugs were then calculated by determining the anti-log of the derived drug’s concentration value (in log_10_) from 50% of cell viability.

### 2.7. Chemo-Sensitization of CD133+ CSCs to MSC-TRAIL

The effect of sensitization in NSCLC-derived CD133+ CSCs using chemotherapies targeting MSC-TRAIL was assessed by treating the sorted (CD133+ and CD133−) and unsorted NSCLC cell lines (A549, H460 and H2170) with the chemotherapeutic drugs (cisplatin, 5-FU and vinorelbine) first, based on the calculated IC_50_ value, for 24 h. Briefly, the NSCLC cell lines were seeded in a 24-well plate (4.0 × 10^5^ cells) in 500 µL of medium containing the chemotherapeutic drugs according to their IC_50_ values. The next day, sensitized cells were harvested and re-seeded again (1.0 × 10^4^ cells in 50 µL complete medium) with either MSC-TRAIL (1:1 ratio) or rhTRAIL (IC_50_ value of each cell: 12.6 ng/mL for H2170, 218 ng/mL for H460 and 500 ng/mL for A549) in a 96-well plate for another 24 h. The NSCLC cells without any treatment (either chemotherapies, MSC-TRAIL or rhTRAIL) were used as a control for the experiment. The analysis of cell viability/proliferation assay was performed the next day using either the luciferase assay for both A549 and H460 (both cells express luciferase constitutively), or MTS for the H2170 cell line (which does not express luciferase). For the luciferase assay, 600 µg/mL of D-luciferin was added into each well containing the cells and subjected to a bioluminescence reading. For the proliferation assay, 10 µL of MTS (CellTiter 96^®^ Aqueous One Solution Cell Proliferation Assay©; Promega Corporation, Madison, WI, USA) was added into each of the wells. The samples were then incubated for 4 h and subjected to absorbance reading at 490 nm using a plate reader (Envision, Perkin Elmer, Waltham, MA, USA).

### 2.8. Statistical Analysis

Data are presented as means ± standard deviation (SD) of three independent experiments. Comparisons between two groups were performed using the two-tailed *t*-test with *p* < 0.05 considered statistically significant. Analyses were performed using Excel 2010, version 14.0 (Microsoft Corporation, Redmond, WA, USA).

## 3. Results

### 3.1. Population Doublings of MSCs and NSCLC

The proliferation rate between MSCs (MSC-WT, MSC-EV and MSC-TRAIL) versus NSCLC cell lines (A549, H460 and H2170) was evaluated by analysing the cumulative population doublings (CPD) for each cell type (Figure 1A). As indicated in Figure 1B, the NSCLC cell lines were highly proliferative, with PDTs of 19 ± 2.0, 25 ± 5.0 and 23 ± 2.3 h for the A549, H460 and H2170 cell lines, respectively, whereas the MSCs (MSC-WT, MSC-EV, MSC-TRAIL) showed slightly longer PDTs (45 ± 18.6, 48 ± 17.3 and 66 ± 19.5 h).

### 3.2. Chemo-Sensitivity of MSCs versus NSCLC Cell Lines

To compare the chemo-sensitivity of NSCLC cell lines (A549, H460 and H2170) versus MSC variants (MSC-WT/wild-type, MSC-EV/empty-vector, MSC-TRAIL), cells were treated with serially diluted concentration of either cisplatin, 5-FU or vinorelbine for 48 h. At the end of the 2 days of treatment, MTS was added into each well and the plates were subjected to an absorbance reading at 490 nm. A distinct separation in terms of cell viability between MSC-TRAIL and NSCLC cell lines was detected when the cells were treated with cisplatin at concentrations lower than 40 µM, indicating that treatments higher than 40 µM were highly toxic to all cells, including the MSCs (Figure 2A). Significantly higher cell viability was observed in MSC-TRAIL when compared to the NSCLC cells lines for all the 5-FU dosages, suggesting that MSC-TRAIL was less sensitive to the 5-FU treatment (Figure 2B). Compared to the NSCLC cell lines, MSC-TRAIL was also resistant to vinorelbine at concentrations lower than 40 µM (Figure 2C). Furthermore, higher cell viability in the MSC variant (MSC-WT and MSC-EV) was observed compared to the NSCLC cell lines (A549, H2170 and H460) following treatments with the chemotherapies ((cisplatin and vinorelbine; ≤40 μM) and 5-FU; ≤20 mM). This indicates that the MSCs were less sensitive to all three chemotherapies compared to the NSCLC cell lines (Figure 2).

### 3.3. IC_50_ Values of Chemotherapies in NSCLC and MSCs

The IC_50_ values of cisplatin, 5-FU and vinorelbine in the NSCLC cells lines were derived from the chemo-sensitivity assays (Figure 2), before the downstream analysis of chemo-sensitizing in NSCLC cell lines was performed. Treatments with all three chemotherapies yielded higher IC_50_ values in the MSC variants as compared to the NSCLC cell lines (Table 1). The sensitivity to both cisplatin and 5-FU was high in H460 cell line, indicated by lower IC_50_ values (1.5 ± 0.1 µM; cisplatin and 1.1 ± 0.2 mM; 5-FU), compared to both A549 (26.0 ± 1.0 µM; cisplatin and 1.6 ± 0.2 mM; 5-FU) and H2170 (10.6 ± 1.5 µM; cisplatin and 1.5 ± 0.1 mM; 5-FU), respectively. The IC_50_ values of all three chemotherapies in the MSC variants and NSCLC cell lines are simplified in Table 1.

### 3.4. Chemo-Sensitization of A549-Derived CD133+ CSCs

The chemo-sensitizing effects of either cisplatin, 5-FU or vinorelbine to enhance the cytotoxic effect of MSC-TRAIL on A549-derived CD133+ CSCs was evaluated. The unsorted CD133+ and CD133− cells were treated with the chemotherapies (according to the IC_50_ values) for 24 h. The sensitized CSCs were subsequently reseeded and then exposed to MSC-TRAIL or rhTRAIL for another 24 h. The CD133− cells from the A549 cell line were highly sensitive to both cisplatin and 5-FU, shown by a lower cell viability (45.0% ± 3.7% for cisplatin and 36.2% ± 3.2% for 5-FU) compared to the CD133+ (71.6% ± 6.0% for cisplatin and 60.0% ± 3.1% for 5-FU) populations (Figure 3A,B). However, there were no differences in cell viability between the A549-derived CD133+ and CD133− cells pre-treated with vinorelbine (Figure 3C). The findings also indicate that sensitization of the A549-derived CD133+ CSCs using cisplatin significantly enhanced the cytotoxic effect of MSC-TRAIL by reducing the viability of CD133+ CSCs from 62.9% ± 5.8% (MSC-TRAIL treatment alone) to 34.3% ± 2.0% (cisplatin and MSC-TRAIL) (Figure 3A). The sensitization of A549-derived CD133+ cells using 5-FU was also able to increase the effect of MSC-TRAIL, as indicated by the substantial decrease in the percentage of cell viability of the CD133+ CSCs from 72.1% ± 2.0% (MSC-TRAIL treatment alone) to 28.0% ± 6.8% (5-FU and MSC-TRAIL) (Figure 3B). However, the pre-treatment of the A549-derived CD133+ CSCs using vinorelbine was not able to enhance the cytotoxic effect of MSC-TRAIL, as there were no differences in cell viabilities between the combination of MSC-TRAIL and vinorelbine, versus MSC-TRAIL treatment alone (Figure 3C). The analysis of the efficacy of chemo-sensitization in the CD133− and unsorted A549 cells indicated that both cisplatin and 5-FU, but not vinorelbine, were able to enhance the effect of MSC-TRAIL-induced inhibition to both CD133− and unsorted A549 cells. Pre-treatment using cisplatin or vinorelbine prior to rhTRAIL was observed to have no effect on reducing the viability of A549 cells (CD133+, CD133− and unsorted) as no significant changes in cell viability were detected between the use of rhTRAIL as a single treatment versus its use in combination with any of the two chemotherapies (Figure 3A,C). However, a reduction in cell viability in CD133+, CD133− and unsorted A549 cells was observed between rhTRAIL as a single treatment versus in combination with 5-FU, as depicted in Figure 3B.

### 3.5. Chemo-Sensitization of H460-Derived CD133+ CSCs

Our analyses revealed that the CD133− cells from the H460 cell line were quite sensitive to all three chemotherapies—cisplatin, 5-FU and vinorelbine—as indicated by their lower cell viability (34.5 ± 3.0%; cisplatin, 46.5% ± 3.0%; 5-FU and 43.8% ± 1.8%; vinorelbine) compared to the CD133+ (50.4% ± 3.4%; cisplatin, 100.0% ± 2.9%; 5-FU and 90.0% ± 3.1%; vinorelbine) populations (Figure 4A–C). The results also indicate that the chemo-sensitization using all three chemotherapies prior to MSC-TRAIL treatment effectively killed the CD133+ CSCs, as shown by the reduction in cell viability to 11.2% ± 1.0% (cisplatin and MSC-TRAIL) from 17.2% ± 2.3% (MSC-TRAIL only, Figure 4A), 11.5% ± 3.7% (5-FU and MSC-TRAIL) from 22.6% ± 1.6% (MSC-TRAIL only, Figure 4B) and 15.4% ± 0.4% (Vinorelbine and MSC-TRAIL) from 17.7% ± 0.4% (MSC-TRAIL only, Figure 4C). Moreover, chemo-sensitization using 5-FU or vinorelbine was able to enhance the effect of MSC-TRAIL against the unsorted and CD133− population in the H460 cell line, as indicated by the reduction in cell viability in the combined treatment as compared to MSC-TRAIL treatment alone (Figure 4A–C). Pre-treatment using cisplatin was not able to induce a chemo-sensitizing effect on the CD133+, CD133− or unsorted H460 cells to rhTRAIL-mediated inhibition, as shown in Figure 4A. However, the other two chemotherapies—5-FU and vinorelbine—managed to enhance the cytotoxic effect of rhTRAIL to kill the CD133+, CD133− and unsorted H460 cells (Figure 4A–C).

### 3.6. Chemo-Sensitizing of H2170-Derived CD133+ CSCs

Treatment using either one of the three chemotherapies did not induce changes in the percentage of cell viability between the CD133+ and CD133− cells in the H2170 cell line. There were no significant changes in cell viability in the CD133+ CSCs between MSC-TRAIL used as a single treatment and in combination with cisplatin, as depicted in Figure 5A. However, a significant reduction in H2170-derived CD133+ cell viability was detected when the MSC-TRAIL was used in combination with 5-FU (9.0% ± 2.4%) or vinorelbine (15.7% ± 1.6%) versus MSC-TRAIL used as a single treatment (20.7% ± 1.1% for the 5-FU group, 22.6% ± 1.6% for the vinorelbine group) (Figure 5B,C). Furthermore, all three chemotherapies were also found to enhance the killing efficacy of rhTRAIL on H2170-derived CD133+ (Figure 5A–C). It was also observed that the combinations of rhTRAIL or MSC-TRAIL with the chemotherapies (cisplatin, 5-FU or vinorelbine) effectively reduced the viability of both CD133− and unsorted H2170 cell lines.

## 4. Discussion

Several studies have reported the anti-tumor activity of MSC-TRAIL against lung cancer, specifically in pre-clinical models of NSCLC [48,49,50]. However, the characteristic of TRAIL resistance has been observed in some of the NSCLC cell lines, indicating that not all the NSCLC subtypes would benefit from MSC-TRAIL treatment [51,52]. In this study, we have shown for the first time that chemo-sensitization of the NSCLC cell lines and their CSCs, particularly the CD133+ population, using first-line chemotherapies such as cisplatin, 5-FU and vinorelbine was able to enhance the anti-tumor effect of MSC-TRAIL. Furthermore, chemo-sensitization of A549 and its CSCs is able to reverse the characteristic of TRAIL resistance, leading to the effective killing of the tumor cells.

An effective anti-tumor response between the combination of MSC-TRAIL and tumor chemo-sensitization can be achieved using chemotherapies that have a strong ability to kill tumors without harming the MSC-TRAIL. Studies have suggested that cells with a higher proliferation rate, such as cancer cells, are sensitive to the effect of chemotherapy compared to slower-dividing cells [53]. Therefore, comparing the proliferation activity between NSCLC cells and MSC-TRAIL might indicate how sensitive MSC-TRAIL is to the effect of chemotherapy. The proliferation activity of cells can be analysed using a growth curve and population doubling time (PDT), with a higher PDT indicating cells with a slower proliferation activity and greater chemoresistance [54]. As shown in Figure 1, the MSCs presented higher PDT values compared to those of the NSCLC cell lines, indicating that the MSCs need a longer time to divide. This is consistent with the chemosensitivity assay, illustrated in Figure 2, that showed higher IC_50_ values in the MSC variants compared to the NSCLC cell lines (Table 1), suggesting that the MSCs are less sensitive to the chemotherapies compared to the NSCLC cells. Therefore, the combination of MSC-TRAIL and first-line chemotherapies could result in an excellent anti-cancer effect as the efficacy of MSC-TRAIL is less likely to be affected by the chemotherapies.

Most chemotherapies are unable to effectively target CSCs due to several factors, including an enhanced DNA repair mechanism, high expression of drug-resistant proteins and evasion of apoptosis [55]. Furthermore, the ability of CSCs to become dormant during treatment is also one of the main factors contributing to tumor relapse in most cancers [56]. Our study showed that treatment using cisplatin or 5-FU resulted in a lower cell viability detected in the CD133− population compared to the CD133+ population in the A549 cell line (Figure 3A,B). A similar observation was also noted for the H460 cell line, where treatment using any one of the three chemotherapies yielded a lower cell viability in the CD133− population than in the CD133+ population (Figure 4). These findings indicate that the CD133+ populations in both the A549 and H460 cell lines are less sensitive to the chemotherapies, and therefore exhibit the characteristic of CSCs. Moreover, treatment using cisplatin resulted in higher cell viability in the unsorted H460 cells than in its CD133+ population, suggesting that the heterogeneous population of CD133+ cells and several other CSCs, such as ALDH, CD166+ and CD44+ cells, in unsorted H460 might contribute to this observation (Figure 4A) [57,58]. There were no changes in cell viability detected between the CD133+ and CD133− populations from the H2170 cell line after treatment with all three chemotherapies, indicating that both populations exhibit the same degree of sensitivity to the chemotherapies (Figure 5).

Although our previous study showed that the A549-derived CD133+ CSCs were highly resistant to MSC-TRAIL [47]; pre-treatment of the A549-derived CD133+ population with cisplatin significantly enhanced the cytotoxic effect of MSC-TRAIL in killing the CSCs (Figure 3A). This observation might be due to the ability of cisplatin to sensitize tumor cells to the effect of TRAIL by enhancing DR5 receptor expression [59]. It was also noted that the 5-FU treatment substantially enhanced the effect of MSC-TRAIL and rhTRAIL to inhibit the A549 cell line and its CSCs (Figure 3B). Mutation of the *KRAS* gene in the A549 cell line [60] may contribute to this observation, as indicated by a study that showed the treatment of 5-FU might preferentially sensitize *KRAS*-mutated NSCLC samples to TRAIL-induced apoptosis [61]. Furthermore, pre-treatment using vinorelbine on the CD133+, CD133− and unsorted A549 cells was unable to enhance the effect of MSC-TRAIL and rhTRAIL against these cells. This implies that the vinorelbine IC_50_ value was not effective to induce chemo-sensitization on the A549 cells; and therefore, using a higher concentration of vinorelbine could exert a greater combined effect. Sensitization using cisplatin was unable to enhance the effect of rhTRAIL against the A549 cell line (CD 133+, CD133− and unsorted) as no significant changes in cell viability were detected between the single rhTRAIL treatment and in combination with chemotherapy. This observation might be due to the short half-life of TRAIL, at about 32 min, leading to the ineffective TRAIL concentration during the 24 h of rhTRAIL treatment in regard to the TRAIL-resistant A549 cell line [62].

Chemo-sensitization of H460-derived CD133+ CSCs using either cisplatin, 5-FU or vinorelbine prior to MSC-TRAIL treatment was able to significantly enhance the killing effect of MSC-TRAIL against the CSCs (Figure 4). Pre-treatment using either 5-FU or vinorelbine was also able to increase the effect of rhTRAIL to destroy H460-derived CD133+ cells (Figure 4B,C). Furthermore, chemo-sensitization using 5-FU or vinorelbine enhanced the effect of MSC-TRAIL and rhTRAIL to kill both CD133− and unsorted H460 cells, suggesting that the combination treatment was effective in killing not only the CD133+ CSCs, but the other populations as well. However, pre-treatment using cisplatin prior to rhTRAIL in the CD133+ population was noted to slightly reduce the viability of the CSC compared to the rhTRAIL treatment alone, which indicates that a higher concentration of rhTRAIL or cisplatin is needed to induce a stronger combined effect to target the CSCs (Figure 4A).

The efficacy of rhTRAIL in targeting H2170-derived CD133+ cells was improved through cisplatin chemo-sensitization (Figure 5A). However, pre-treatment using cisplatin on H2170-derived CD133+ cells did not enhance the effect of MSC-TRAIL in killing CSCs as no changes in cell viability were detected between the MSC-TRAIL treatment and its use in combination with cisplatin. Since a study has reported a strong chemo-sensitization effect of using cisplatin at 72 h of exposure, we postulate that the sensitivity of H2170-derived CD133+ cells to MSC-TRAIL could be improved by increasing the time of exposure of the CSCs to cisplatin [63]. Findings from this study also demonstrated that 5-FU and vinorelbine were an effective chemo-sensitizer, used to enhance the effects of both MSC-TRAIL and rhTRAIL against the H2170 cell line, particularly the CD133+ CSCs (Figure 5B,C). Pre-treatment using 5-FU or vinorelbine was also noted to increase the effect of both MSC-TRAIL and rhTRAIL against the CD133− and unsorted H2170 cells, suggesting that the combination of either 5-FU or vinorelbine with MSC-TRAIL is an effective treatment against the H2170 cell line and its CSCs. However, to verify this observation, analysis of cleaved caspase-3 can be performed to confirm the induction of apoptosis in the CSCs by MSC-TRAIL.

We have demonstrated that a pre-treatment with first-line chemotherapies was able to enhance the sensitivity of the NSCLC-derived CD133+ cells to the effects of MSC-TRAIL. However, the exact mechanism leading to this observation remains to be elucidated. One particular reason might be due to the ability of these chemotherapies to regulate the expression of cellular proteins, which may subsequently enhance the transcription of TRAIL receptors, such as DR4 and DR5, leading to the greater sensitivity of tumor cells and CSCs to TRAIL. For example, a pre-treatment using cisplatin has been reported to down-regulate FADD-like IL-1β-converting enzyme (c-FLIP) activity, which is an anti- apoptotic protein, leading to an increase in DR5 receptor expression in glioma-derived CSCs [64]. On the other hand, chemo-sensitization using 5-FU has been shown to upregulate the expression of caspases and DR5 in a model of adenocarcinoma, resulting in the activation of both the intrinsic and extrinsic apoptosis pathways [65]. Pre-treatment using 5-FU has also been reported to increase the expression of Bcl-2-associated X (Bax) protein in A549 cells, leading to a greater sensitivity of the tumor cells to TRAIL [66]. However, to confirm that the enhancement in TRAIL sensitivity in the CSCs after chemo-sensitization was indeed due to the increase in the expression of TRAIL receptors, analysis of the expression of DR4 and DR5 in the CSCs after chemo-sensitization can be performed in future studies.

## 5. Conclusions

This study has demonstrated that the sensitizing of NSCLC cell lines and their CSCs, specifically the TRAIL-resistant A549 cell line, using first-line chemotherapies could enhance the effect of MSC-TRAIL against NSCLC. However, different chemotherapies may have different chemo-sensitizing effects on the NSCLC subtype. Therefore, selecting chemotherapies for the right NSCLC subtypes is crucial for a meaningful combination effect. Despite the promising outcomes from this in vitro study, we are uncertain if these observations could be replicated in vivo. Therefore, we intend to conduct further studies on the lung cancer mouse model in order to verify the efficacy of this strategy. The favorable results from this study suggest an alternative approach of using MSC-TRAIL as a complement for the treatment of NSCLC patients. However, to achieve optimal results, the chemotherapies should have no effect on the MSCs, and this can be assured by choosing chemotherapies that specifically target the tumor without harming MSC-TRAIL.

## Figures and Tables

**Figure 1 biology-10-01103-f001:**
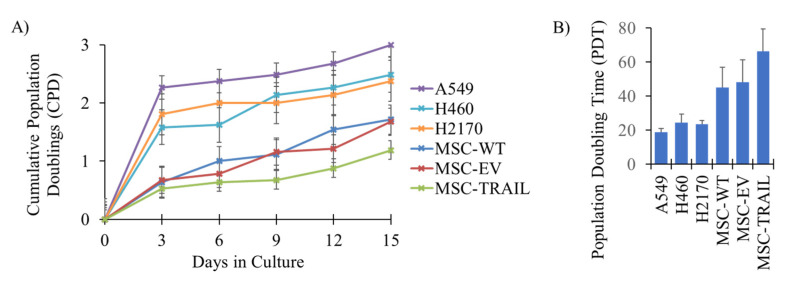
The cumulative population doublings (CPD) and population doubling time (PDT) of NSCLC (A549, H460 and H2170) and MSCs ((MSC-WT (wild-type), MSC-EV (empty vector) and MSC-TRAIL). Cultured cells were harvested every 72 h and their CPD and PDT were calculated. (**A**) CPD presented in the graph indicates the proliferation rate of each cell line at different days in culture. (**B**) Based on the CPD, the population doubling time (PDT) was calculated in hours for each of the cell lines.

**Figure 2 biology-10-01103-f002:**
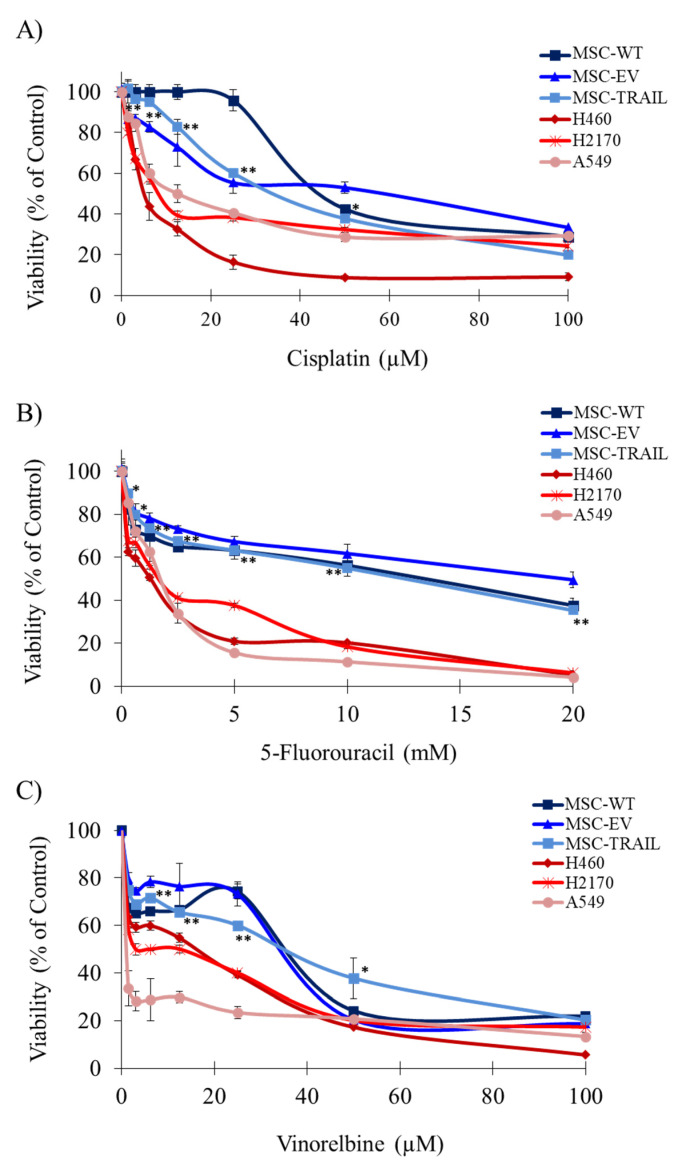
The cell viability of NSCLC cell lines (H460, H2170 and A549) and MSC variants (MSC-WT, MSC-EV and MSC-TRAIL) treated with chemotherapies. (**A**) Higher cell viability in all MSC variants compared to the NSCLC cell lines treated with cisplatin at concentrations lower than 40 µM. (**B**) A distinct separation in terms of cell viability was observed between NSCLC cell lines and the MSC variants at different 5-FU dosages. (**C**) Compared to the NSCLC cell lines, MSC variants were resistant to the vinorelbine treatment at concentrations lower than 40 µM. However, as the concentration increases to 100 µM, the cytotoxicity of vinorelbine becomes more prominent for all cells. ((* *p* < 0.01, ** *p* < 0.001; *t*-test (MSC-TRAIL vs NSCLC cell lines), *n* = 3)). WT, wild-type; EV, empty vector; 5-FU, 5-fluorouracil.

**Figure 3 biology-10-01103-f003:**
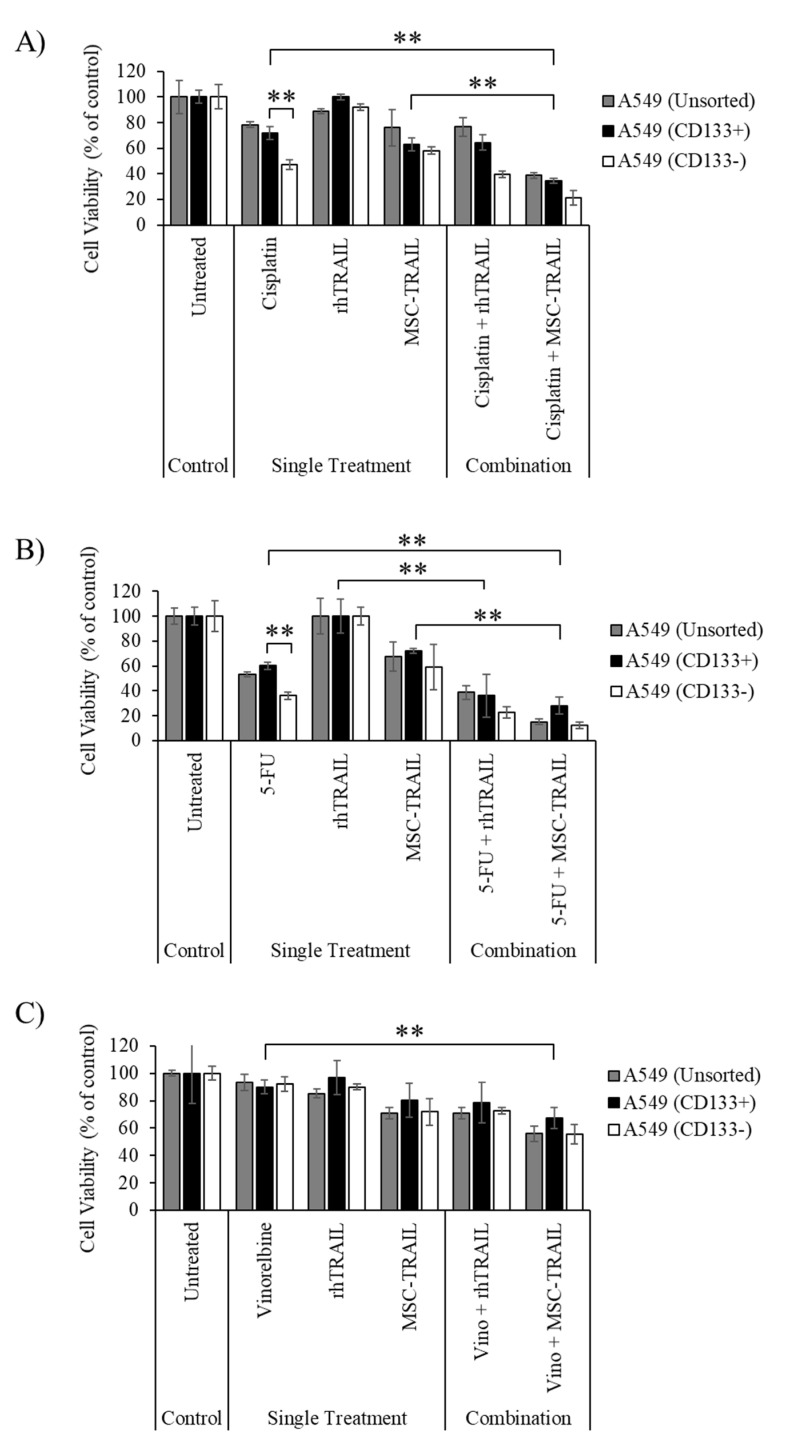
The chemo-sensitization effect in A549-derived CD133+ CSCs to MSC-TRAIL. (**A**,**B**) Chemo-sensitization of A549-derived CD133+ CSCs using either cisplatin or 5-FU significantly increased the effect of MSC-TRAIL in killing CSC. (**C**) The combination between MSC-TRAIL and vinorelbine appears to have no significant effect in enhancing the inhibition of CSCs when compared to MSC-TRAIL alone, as no changes in cell viability were detected between the two groups (** *p* < 0.001; *t*-test, *n* = 3). 5-FU, 5-fluorouracil; Vino, vinorelbine.

**Figure 4 biology-10-01103-f004:**
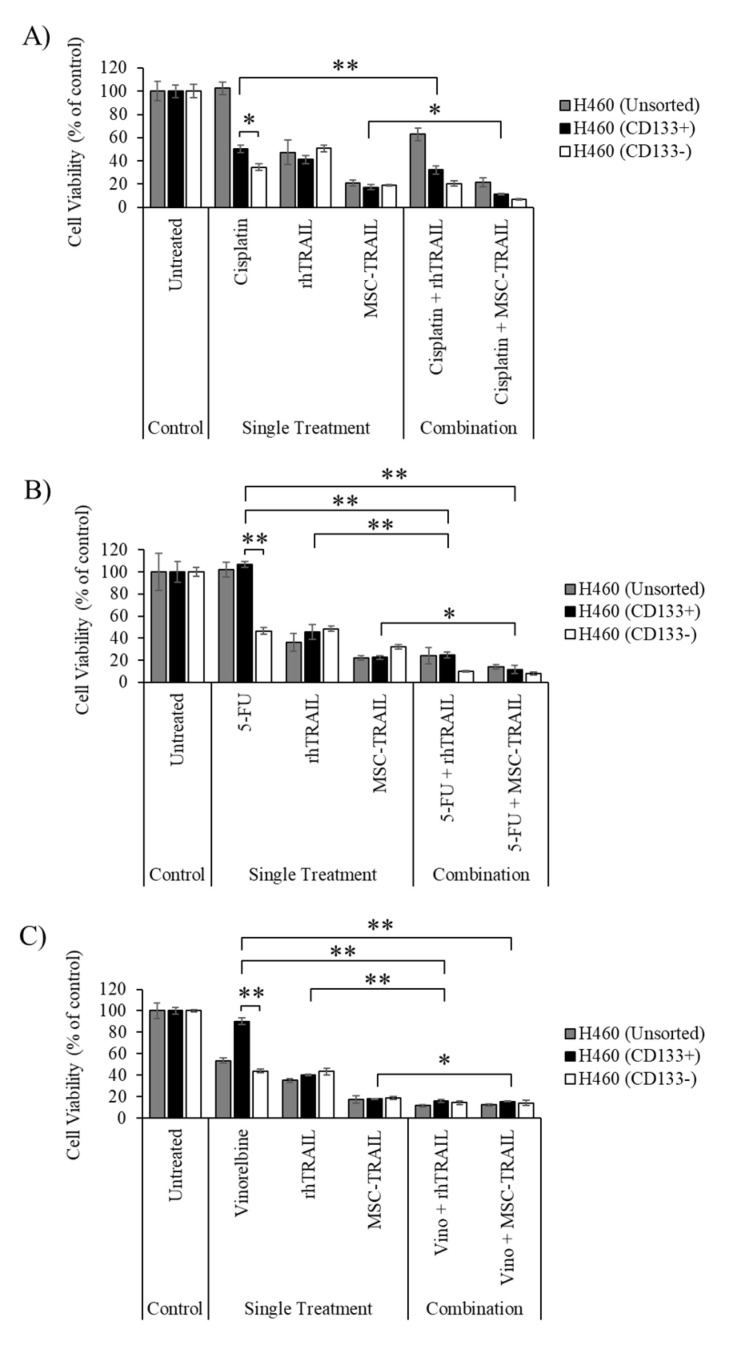
Cell viability of chemo-sensitized H460-derived −+ CSCs to MSC-TRAIL treatment. (**A**) Chemo-sensitization using cisplatin enhanced the cytotoxic effect of MSC-TRAIL, but not rhTRAIL in the H460-derived CSCs. (**B**,**C**) Chemo-sensitization of H460-derived CSCs using 5-FU or vinorelbine/vino increased the cytotoxic activity of both MSC-TRAIL and rhTRAIL (** *p* < 0.001, * *p* < 0.01; *t*-test, *n* = 3). 5-FU, 5-fluorouracil; Vino, vinorelbine.

**Figure 5 biology-10-01103-f005:**
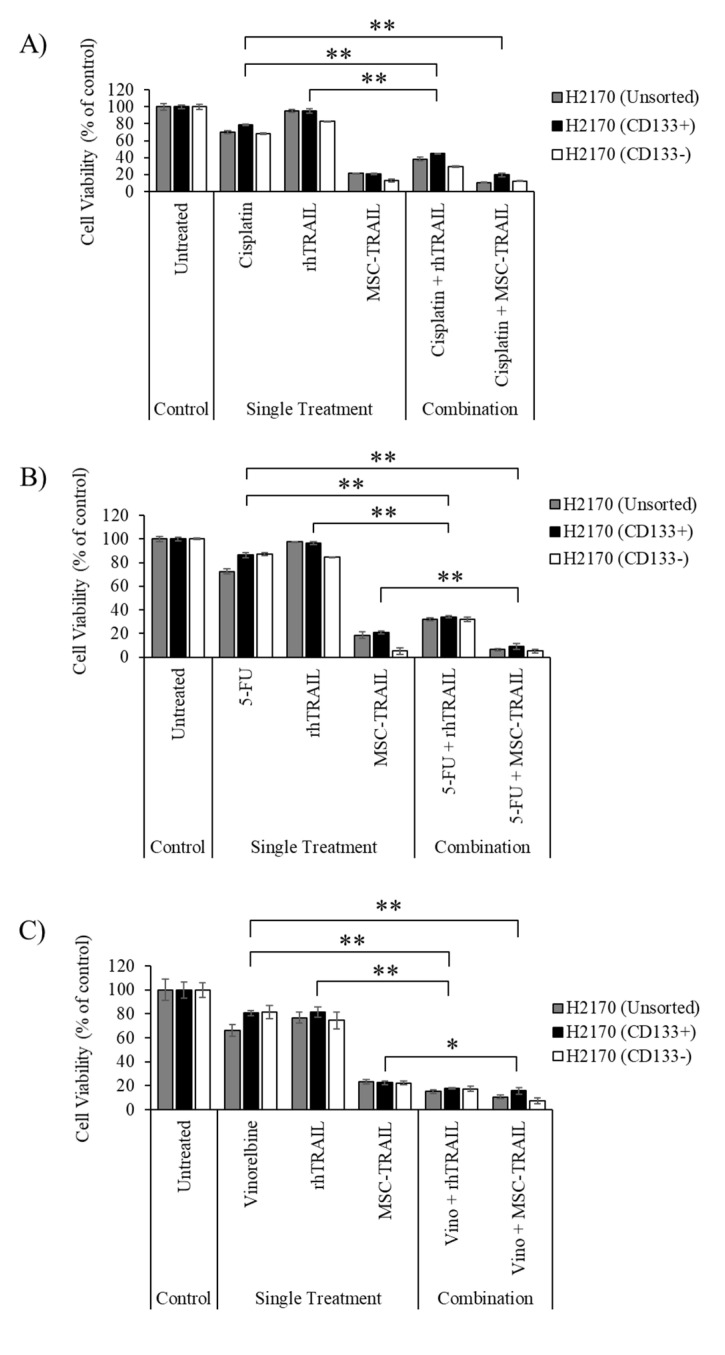
The effect of chemotherapy pre-treatment on the efficacy of MSC-TRAIL in H2170-derived CD133+ cells. (**A**) The pre-treatment using cisplatin was not able to enhance the effect of MSC-TRAIL in H2170-derived CD133+ CSCs. (**B**,**C**) Pre-treatment using either 5-FU or vinorelbine/vino was able to increase the cytotoxic effect of both MSC-TRAIL and rhTRAIL to kill the CSCs (** *p* < 0.001, * *p* < 0.01; *t*-test, *n* = 3). 5-FU, 5-fluorouracil; Vino, vinorelbine.

**Table 1 biology-10-01103-t001:** The IC_50_ values of all three chemotherapies in MSC variants and NSCLC cell lines.

Cell Types	Cisplatin (µM)	5-FU (mM)	Vinorelbine (µM)
MSC-WT	73.3 ± 5.8	13.1 ± 1.8	36.3 ± 5.5
MSC-EV	40.3 ± 8.4	26.3 ± 6.7	36.0 ± 1.0
MSC-TRAIL	29.0 ± 2.6	10.0 ± 1.0	32.3 ± 3.21
A549	26.0 ± 1.0	1.6 ± 0.2	<1.5
H2170	10.6 ± 1.5	1.5 ± 0.1	10.67 ± 1.5
H460	1.5 ± 0.1	1.1 ± 0.2	8.57 ± 1.5

Results are the mean ± standard deviation (SD).

## Data Availability

Not applicable.

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
