# Peer review of "Chemo-Sensitization of CD133+ Cancer Stem Cell Enhances the Effect of Mesenchymal Stem Cell Expressing TRAIL in Non-Small Cell Lung Cancer Cell Lines"

_biology, 2021, doi:10.3390/biology10111103_

Round 1

Reviewer 1 Report

Kamal Shaik Fakiruddin et al., Anti-tumor activity of MSC-TRAIL can be utilized against TRAIL-resistant tumor after chemo-sensitization of CD133+ cancer stem cells with cisplatin, 5-Fluorouracil and vinorelbine.    

-The manuscript is well written.

-Mostly easy to follow and the figures are well put together.

-Using different cell lines supports authors point. 

-The quality of figures and table presented by authors are satisfactory.

However, there are few issues that in my opinion preclude publication of the manuscript in its present form.

1) Did author try combination of Cisplatin+5-FU+Vino (or different combinations) for chemo-sensitization of CD133+ cancer stem cells before treatment with MSC-TRAIL?

2) Although, author has mentioned brief points during discussion, Manuscript is missing mechanistic details that how chemo-sensitization with given reagent makes cells more sensitive against MSC-TRAIL?

3) What is authors opinion on metabolic state of cells before and after chemo-sensitization?

4) What is the status of cleaved Caspase-3 in cell viability experiments?

5) What is author's view on using mice model for this study?

Reviewer 2 Report

Overall this is a very well written paper and the data is significant and clearly presented.  The concept of chemo-sensitization to improve cellular therapy is clinically relevant and raises the possibility of the progression of these findings into clinical trials. 

Some concerns are that the differences in sensitivity to the three chemotherapies between the MSC-TRAIL and the control, MSC-EV, are somewhat small and it is not clear why the MSC-EV has the observed effect in the Cisplatin treatment group.  It is possible that mere process of lentiviral infection has a sensitizing affect through innate immune sensing pathways. Testing the gene transfer through non-viral methods, such as plasmid transfection, would be an interesting control.

Nonetheless, in current form it is a very well written and concise paper with results that will inspire further experimentation into these pathways.   

Round 2

Reviewer 1 Report

The authors have addressed my concerns satisfactorily.

I recommend publication now.